# Bacterial Decontamination of Water-Containing Objects Using Piezoelectric Direct Discharge Plasma and Plasma Jet

**DOI:** 10.3390/biom14020181

**Published:** 2024-02-02

**Authors:** Evgeny M. Konchekov, Victoria V. Gudkova, Dmitriy E. Burmistrov, Aleksandra S. Konkova, Maria A. Zimina, Mariam D. Khatueva, Vlada A. Polyakova, Alexandra A. Stepanenko, Tatyana I. Pavlik, Valentin D. Borzosekov, Dmitry V. Malakhov, Leonid V. Kolik, Namik Gusein-zade, Sergey V. Gudkov

**Affiliations:** 1Prokhorov General Physics Institute of the Russian Academy of Sciences, 119991 Moscow, Russia; gudkova-vi@fpl.gpi.ru (V.V.G.); dmitriiburmistroff@gmail.com (D.E.B.); ngus@mail.ru (N.G.-z.); s_makariy@rambler.ru (S.V.G.); 2Institute of Physical Research and Technology, Peoples’ Friendship University of Russia (RUDN University), 117198 Moscow, Russia; 3Center for Precision Genome Editing and Genetic Technologies for Biomedicine, Institute of Gene Biology, Russian Academy of Sciences, 119334 Moscow, Russia

**Keywords:** cold atmospheric plasma, plasma-treated solution, reactive oxygen species, reactive nitrogen species, plasma medicine, plasma agriculture, plasma food processing

## Abstract

Cold atmospheric plasma has become a widespread tool in bacterial decontamination, harnessing reactive oxygen and nitrogen species to neutralize bacteria on surfaces and in the air. This technology is often employed in healthcare, food processing, water treatment, etc. One of the most energy-efficient and universal methods for creating cold atmospheric plasma is the initiation of a piezoelectric direct discharge. The article presents a study of the bactericidal effect of piezoelectric direct discharge plasma generated using the multifunctional source “CAPKO”. This device allows for the modification of the method of plasma generation “on the fly” by replacing a unit (cap) on the working device. The results of the generation of reactive oxygen and nitrogen species in a buffer solution in the modes of direct discharge in air and a plasma jet with an argon flow are presented. The bactericidal effect of these types of plasma against the bacteria *E. coli* BL21 (DE3) was studied. The issues of scaling the treatment technique are considered.

## 1. Introduction

Physical plasma is a fully or partially ionized matter that has the properties of quasi-neutrality and collectivity. The variety of methods for creating plasma and the ability to vary its parameters over a wide range determine the distribution of plasma in a large number of technological fields [1]. Thus, physical plasma is already actively used in the modification of surfaces and liquids [2,3,4], the purification of gas and liquid media [5,6,7,8], the synthesis of micro- and nanomaterials [9], medicine [10,11,12,13,14], agriculture [15,16,17,18,19,20], food processing [21,22], etc., and is also planned for use in the future in areas such as energy industry (controlled fusion), space agriculture, and space medicine [23].

In solving biologically related problems, cold atmospheric plasma (CAP) is used, which is characterized by minimized heat transfer to the object and thermal damage. This is primarily because, in such a plasma, most of the particles (ions, electrons, and neutrals) are not in thermal equilibrium. While the bulk of the gas might maintain a relatively low temperature, certain particles within the plasma, particularly electrons, can possess higher energies. They might contribute significantly to chemical reactions. The degree of gas ionization in CAP can have very small values (fractions of a percent), which, from a strictly physical point of view, does not always justify the use of the term “plasma”; however, such a flexible interpretation of the term “cold atmospheric plasma” has been established in the biological and medical sciences.

The effect of CAP on biological objects includes the influence of factors such as reactive oxygen and nitrogen species (ROS and RNS) [24,25], including radicals and ions, electron flow, electric field, ultraviolet radiation, and heating. Reactive species play a key role, and they can induce oxidative stress, disrupting the cell membrane and causing damage to proteins and nucleic acids. This targeted damage can lead to the inactivation of microorganisms, including bacteria, viruses, and fungi. Furthermore, CAP has been found to modulate cellular processes. It can influence cell signaling pathways, leading to alterations in gene expression and cell behavior [26,27]. This capability led to therapeutic applications, such as in wound healing and cancer treatment [28,29,30].

In the agricultural sector, CAP technology has shown potential for pest control and seed germination enhancement, so it offers an environmentally friendly alternative to traditional pesticides and growth stimulants. Additionally, CAP treatments have been explored for decontamination and preservation in the food industry. The antimicrobial properties can reduce the microbial load on food surfaces, extending shelf life and enhancing food safety without the use of chemical additives.

CAP treatment methods can be divided into two categories: direct and indirect. In the case of direct treatment, the plasma (or electrical discharge) directly interacts with the surface of the object, which in turn acts as an opposing electrode. In indirect plasma treatments, a discharge is created between electrodes located away from the object, and active particles are delivered to the surface of the object via diffusion or convection or an electric field applied.

Often, indirect plasma exposure also means exposure to biological objects using plasma-treated water or solutions (PTW and PTS) [31,32]. That is, the liquid medium is treated and then applied to the biological object. In this case, of all the factors of plasma influence (ROS, RNS, UV, heat, electrons, and current), only ROS and RNS influence the object. However, research shows that the use of PTW and PTS can often achieve results similar to direct plasma treatment. In addition, this type of treatment is safer for biological objects in terms of potential thermal destruction and is easier to scale.

One of the interesting and promising types of direct treatment is processing using direct piezoelectric discharge (PDD) plasma [33,34]. Plasma sources using PDD are primarily characterized by energy efficiency, compactness, and flexibility in the design of the working (output) device [35], which directly acts as a high-voltage electrode that creates plasma. The use of such plasma sources has already proven itself in medicine, where targeted effects are required, for example, on a tumor (cancer cells).

A series of studies using the “CAPKO” plasma source showed that PDD has a dose-dependent cytotoxic effect on cancer cells in vitro. At the same time, the antitumor drug doxorubicin enhances this effect, and the antioxidant dihydroquercetin weakens it [36]. The combined effect of PDD and doxorubicin on tumor cells showed an increase in the effectiveness of doxorubicin by 40%. The observed increase in the activity of the antitumor drug is explained by the formation under the influence of plasma of ROS and RNS, which have additional genotoxic and DNA-damaging effects. Subsequent studies showed that it is possible to use PDD-treated saline as a nonspecific cyto- and genotoxic agent with prolonged effects [37]. With increasing time of solution treatment with PDD, an increase in autophagy activation is observed in three types of cells: mononuclear leukocytes from apparently healthy donors, myeloid leukemia cell line K562, and T-lymphoblastic leukemia cell line Jurkat. It was also shown that a chemically prepared mixture of nitrites, nitrates, and hydrogen peroxide in concentrations similar to PTS has an effect on cell culture that is fundamentally different from the effect of PTS [38]. A chemically prepared solution is more cyto- and genotoxic and causes necrosis, while under the influence of PTS, apoptotic processes slowly start in the cells.

The anticancer properties of PTS in combination with drugs are one of the most popular topics in modern plasma medicine. The study [39] compares the effects of four PTS prepared using PDD from isotonic solutions and also studies their combined cytotoxic effect with doxorubicin and medroxyprogesterone acetate (MPA). Analysis of the influence of the studied drugs on the formation of radicals in the incubation medium, the viability of K562 myeloid leukemia cells, and the processes of autophagy and apoptosis in them revealed two key conclusions. First, when using PTS with doxorubecin in cancer cells, autophagy is the predominant process. Secondly, the combination of PTS with MPA enhances apoptotic processes.

The “CAPKO” source in the DBD plasma generation mode was studied in relation to the effect on the quality of growth of pear and cherry after grafting [40,41,42]. Rootstock and scion cuts were treated with DBD CAP during grafting. This led to better formation of the vascular system and faster growth of trees.

Despite the results of using PDD in the treatment of liquids, animal cells, and plant surfaces presented in the literature, the results of antibacterial studies are poorly presented [43,44]. In this article, we set out to study the capabilities of the PDD and PDD-driven argon plasma jet [45,46] in solving problems of bacterial decontamination of the surface of biological objects, that is, water-containing objects, in combination with the issue of scaling the technology. 

## 2. Materials and Methods

### 2.1. Plasma Source and Treatment Methods

The multifunctional device “CAPKO” (GPI RAS, Moscow, Russia) was used as a plasma source. The operating device of this source allows us to change the type of plasma treatment on the fly by selecting a suitable replaceable unit (Figure 1). There are four operating modes.

The first and second modes are corona-like discharges with a plasma bridge and PDD with the optional ability to operate in various gases (Figure 2). In this case, the replaceable unit is a ring made of tempered glass with a lock on the body. The end of the piezotransformer (PT) acts as a high-voltage electrode. The processing object acts as a response electrode. When the distance between the electrodes is more than 5 mm (may vary depending on the electrical characteristics of the treated object), the plasma generated by the source is a corona discharge. At distances of about 5 mm or less, a spark channel is formed, which is commonly called PDD. In this processing mode, the spark exists for several nanoseconds and is repeated in each half-cycle of the supply voltage, and the position of the channel changes. Thus, despite the rather high temperatures of the particles in the channel, the integral temperature of the object increases slightly. However, a local change in surface temperature in the cross section of the spark channel can lead to point thermal destruction of biological objects and, as a result, the PDD mode is used to create PTW and PTS [46] or when processing biological objects covered with a significant layer of liquid.

The third processing mode is a plasma jet with a flow of inert gas or a mixture of gases (Figure 3). In this case, the replaceable unit is a nozzle with a hole. The distance to the treatment object can vary widely from a few millimeters to a couple of centimeters, which will determine the treatment area and the concentration of generated ROS and RNS. Spark discharges occur in an inert gas environment; however, in this system, local heating of the surface of the object in the cross section of the spark channel, as a rule, does not reach values critical for the biological object. Thus, this mode is gentler compared to PDD.

The fourth processing mode is dielectric barrier discharge (DBD) [40]. The replaceable unit is a holder with a silicone insert. The object is processed by contact. CAP occurs in a narrow air gap when a silicone insert comes into contact with a biological object. This mode is the safest for biological applications and is easier to scale; however, due to contact, it is not very effective when processing liquids and surfaces with complex topography.

The piezotransformer (Elpa Research Institute, Moscow, Russia) operated at half resonant frequency: 21.0 ± 0.5 kHz. The exact value varies depending on the electrical characteristics of the electrode–gas–object system. With an RMS input voltage of 60 V, the RMS voltage at the output of the piezotransformer acting as the active electrode is ~6 kV. 

The experiments were carried out in PDD mode in ambient air and in argon plasma jet (Ar-PJ) mode with argon flow 2 L/min. PDD is the most effective mode in terms of the rate of generation of ROS and RNS in the liquid, which, in turn, are the most significant biologically active agents. The Ar-PJ mode allows us to process biological objects directly with a lower probability of spot thermal damage to the surface compared to PDD. The degree of exposure was varied by changing the duration of treatment.

The power consumption of the piezotransformer did not change in the experiments in all modes and amounted to 3 W. The average power supplied to the discharge was approximately 1.5 W. This power is distributed during the processing between all spark discharges with a duration of ~10 ns, followed by a frequency of ~42 kHz (approximately one spark per half-cycle of the supply voltage). In the Ar-PJ mode, due to the use of an argon flow, processing is possible with a larger distance between the electrode and the object. In addition, the gas flow cools the surface. This significantly affects the chemical reactions occurring in the system. In the previous article [45], we present the results of preliminary measurements of electron and ion temperatures in various modes for the “CAPKO” plasma source.

We used two models of water-containing objects with bacterial contamination. The first of these is a suspension of phosphate-buffered saline (PBS) with bacteria. This model allows us to evaluate the contribution of ROS and RNS to the bactericidal activity of plasma treatment. The second model is a thin layer of bacteria on the surface of the agar. In this case, the effective area of plasma treatment was studied, which is important for solving the issue of scaling the technology.

To evaluate the changes in the treated surface temperature, we used an Optris PI 640 infrared camera (Optris, GmbH, Berlin, Germany) and Optris PIX Connect software (3.15.3090.0, Optris, GmbH, Berlin, Germany).

All treatments (Table 1) had three replicates which were carried out under constant environmental conditions and the same experimental protocols.

### 2.2. Methods for Determination of the Concentration of Chemical Compounds in Liquid

The concentrations of hydrogen peroxide and nitrite ions were determined spectrophotometrically using the FOX assay and Griess assay [47,48,49] by determining the optical density of solutions at a wavelength of 560 nm (waiting time 5 min) and 525 nm (waiting time 20 min), respectively. For this purpose, a HACH LANGE DR-5000 spectrophotometer (HACH LANGE GmbH, Düsseldorf, Germany) was used. The length of the absorbing layer (cuvette edge) was 1 cm. NO3− ions were detected using LAQUAtwin NO3-11 (HORIBA Advanced Techno, Kyoto, Japan). The conductivity, pH, and redox potential of the liquids were determined using a SevenExcellence multichannel meter (Mettler Toledo, Greifensee, Switzerland).

The name of the reagent FOX stands for “ferrous oxidation in xylenol orange”. The method is based on the oxidation of iron ions Fe^2^+ in Mohr’s salts under the action of hydrogen peroxide to Fe^3^+ in an acidic environment with the formation of a colored complex. Diagnostics refers to indirect methods. Calibration of the counting solution was carried out using 37% H_2_O_2_. The method is highly selective; in addition, the authors were convinced that there was no influence of nitrite ions on the measurements of hydrogen peroxide. During measurements and calibration, the test sample and the FOX reagent were mixed 1:1.

The Griess assay is based on the ability of nitrite ions to diazotize sulfanilic acid to form a red-violet azo dye, a diazo compound, as a result of reaction with α-naphthylamine (nitration of the dye). Calibration was carried out by measuring the optical densities of a NaNO_2_ solution at various concentrations. During measurements and calibration, the test sample and the Griess 2:0.15 reagent (2 mL and 150 μL) were mixed.

### 2.3. Bacterial Culture and Cultivation Protocol

*E. coli* BL21 (DE3) was used in all experiments. All manipulations with bacterial cell cultures were carried out in a “Laminar-S” class II biological safety cabinet (Lamsystems, Miass, Russia) in compliance with standard aseptic measures. Freshly prepared Lysogeny broth (LB) according to Miller (GRiSP, Porto, Portugal), sterilized by autoclaving at 121 °C for 15 min, and agar was used as a medium for the cultivation of bacterial cells *E. coli* BL21 (DE3) (Muller-Hinton II (State Scientific Center for Medical and Biology, Serpukhov, Moscow region, Russia). An inoculated PBS solution without plasma treatment diluted with sterile LB broth (1:1) and a sterile PBS solution diluted with LB broth (1:1) were used as positive and negative controls, respectively.

### 2.4. Protocol for the Treatment of Bacterial Suspension Cultures

An overnight suspension culture of *E. coli* BL21 (DE3) containing ~10^8^ cells was diluted in a sterile PBS solution (Sigma, St. Louis, MO, USA) to a final concentration of ~10^6^. For experiments on the treatment of bacterial cell suspensions, a 50 mL PBS solution inoculated with *E. coli* cells was prepared by adding 50 µL of a bacterial cell suspension at a concentration of 10^6^ CFU/mL. Treatment of 1 mL of solution was carried out in 6-well plates (TPP, Trasadingen, Switzerland). The distance from the plasma source to the surface of the solution in the cup was constant and amounted to 10 mm for Ar-PJ and 5 mm for PDD. After treatment, 1 mL of sterile LB broth was added to each well; the plate was closed and placed in an ES-20 incubator shaker (Biosan, Riga, Latvia). Cultivation was carried out for 18 h at 37 °C and 150 rpm. At the end of cultivation, the optical density of cell suspension cultures (V = 300 μL) was recorded at a wavelength of 600 nm (OD600), placed in the wells of 96-well transparent polystyrene plates (Corning, Glendale, AZ, USA). Optical density was recorded at a wavelength of 600 nm using an Allsheng Feyond-400 tablet spectrophotometer (Allsheng, Hangzhou, China).

### 2.5. Protocol for the Treatment of Bacterial Cells on Agar

For experiments on the direct effect of plasma treatment on *E. coli* cells localized on agar, an inoculum of an *E. coli* bacterial culture was prepared in PBS (Sigma, USA), containing about 10^6^ CFU/mL. To study the effect of plasma treatment of bacteria on the surface of the agar, 100 μL of the resulting inoculum was evenly distributed over the surface of a sterile Petri dish (90 mm) containing Mueller-Hinton II agar (State Scientific Center for Medical Biology, Russia). After processing, the Petri dishes were closed with a lid and placed upside down in the incubator. Cultivation was carried out for 24 h at 37 °C. The size of inhibition zones and the percentage of inhibition from the total surface area of the seeded Petri dish were recorded using ImageJ2 (Fiji) software (NIH, Bethesda, MD, USA) from photographs of Petri dishes by highlighting the area of interest.

## 3. Results

### 3.1. Reactive Oxygen and Nitrogen Species Generation in Phosphate-Buffered Saline

H_2_O_2_ and NO_2_− concentrations in 1 mL of PBS treated with PDD and Ar-PJ are presented in Figure 4. In both modes, the dependence on the treatment duration is close to linear. PDD, as expected, has better efficiency in terms of ROS and RNS generation. H_2_O_2_ concentration increased to 49 ± 8 μM in PDD mode and 33 ± 3 μM in Ar-PJ mode. NO_2_− concentration in PDD mode reached 1150 ± 92 μM, while NO_3_− concentration was at the level of ~2000 μM. Ar-PJ treatment shows significantly lower RNS generation, which is limited by nitrogen dissolved in the liquid and its weak diffusion into the discharge region from the ambient air.

### 3.2. Plasma Treatment of Bacterial Culture

Two modes of plasma treatments (PDD and Ar-PJ) were tested against suspension cultures of Gram-negative *E. coli* cells (Figure 5) for 15, 30, 45, and 60 s. Direct plasma treatment for 15 s had no significant effect on suspension bacterial cultures. A significant inhibitory effect was observed with both types of treatment over a 45-s exposure. The 60 s treatment had the most pronounced effect; the optical density of the cell suspension in this group for both types of plasma treatment was comparable with the values of the negative control and amounted to about 0.4 a.u. The difference in processing results for PDD and Ar-PJ modes correlates with differences in H_2_O_2_ generation in PBS.

We also studied the direct effect of the considered types of plasma treatment on bacterial cells on agar. This made it possible to determine the area of effective plasma action in each mode. The treatment duration was chosen to be the shortest of those that showed good efficiency—45 s. Treatment was also carried out for 500 s to monitor the dynamics of the increase in the area of bacterial inhibition and possible areas of damage to the heat-sensitive surface. The data obtained indicate the presence of an inhibitory effect with 45 s of treatment with both PDD and Ar-PJ (Figure 6). The average % inhibition area was 18 and 24 for PDD and Ar-PJ treatment for 45 s, % inhibition area was 5.2 ± 0.25% and 1.45 ± 0.08% for PDD and Ar-PJ, respectively. Prolonged exposure for 300 s also had a pronounced inhibitory effect. The proportion of the inhibition region was 22.8 ± 1.15% and 9.7 ± 0.44% for PDD and Ar-PJ, respectively.

In the Ar-PJ mode, the treatment is more uniform but has a smaller area. In PDD mode, the spark channel changes its position within a much larger range, which, on the one hand, increases the effective treatment area, but on the other hand, is a random process.

The time dependence of the local maximum temperature on the surface of the sample during processing is presented in Figure 7. In the case of processing contaminated PBS, the maximum local surface temperature (hot spot) practically does not increase when processing in the Ar-PJ mode. In PDD mode, it can reach 50 degrees at the point of contact of the discharge with the surface, but the temperature near this point is much lower.

When treating the surface of contaminated agar, the temperature of the hot spot in the Ar-PJ mode is around 30 degrees throughout the entire duration of treatment. In PDD mode it can reach 80 degrees, due to which the agar melts and takes on a different color (this can be seen in Figure 6). However, this temperature is achieved in a region of a small area; the temperature a few millimeters from this point is significantly lower.

## 4. Discussion

The effectiveness of CAP against bacteria is attributed to an interplay of mechanisms that disrupt bacterial cellular structures and functions. One of the most significant factors of CAP’s antibacterial action is the generation of various reactive species, including reactive oxygen species (ROS) and reactive nitrogen species (RNS). These species, such as ozone, hydrogen peroxide, superoxide radicals, hydroxyl radicals, and nitric oxide, induce oxidative stress within bacterial cells. The oxidative stress leads to damage to vital cellular components, including DNA, proteins, and lipids. The specific mechanisms of CAP’s antibacterial effects can vary depending on factors such as the type of bacteria, characteristics of the plasma, and the conditions of exposure [23,50].

For example, the investigation [51] involved assessing the antimicrobial impacts of surface microdischarge sources on clinically relevant Gram-negative and Gram-positive bacteria, along with the fungus *Candida albicans*. The tested method exhibited efficacy against both types of bacteria and the mentioned fungus. Furthermore, the inactivation of spores occurred at a faster rate compared to conventional sterilization techniques.

In the study [43], decontamination experiments were conducted utilizing an atmospheric pressure plasma jet driven by a piezoelectric transformer. The efficacy of the decontamination process was assessed in terms of reduction rates for *Pseudomonas aeruginosa* and *Staphylococcus aureus* bacterial strains. The experiments involved the use of an argon plasma jet operating at atmospheric pressure.

In dynamic simulations [52,53], it was observed that plasma species have the capability to disrupt critical bonds within the cell wall, specifically the peptidoglycan structure, in Gram-positive bacteria. Simultaneously, these plasma species induced membrane lipid peroxidation in Gram-negative bacteria. The consequential disturbance to the outer cell shell results in the leakage of vital cellular components, including potassium, nucleic acids, and proteins. Once the integrity of the cell wall is compromised, reactive species can penetrate the interior of the cell, causing additional damage to DNA and intracellular proteins via oxidative or nitrosative mechanisms [54].

The study [55] revealed a correlation between the inactivation of bacteria by CAP (generated with kINPen [56]) and the thickness of the bacterial cell wall. The findings demonstrate that biofilms formed by Gram-negative species, characterized by a thinner cell wall, exhibit a more rapid inactivation compared to biofilms of Gram-positive bacteria, which possess a thicker cell wall.

It was experimentally revealed [57,58] that ROS plays a key role in inhibiting the growth of Gram-negative bacteria, such as *E. coli*. They act primarily on the cell membrane and cause irreversible damage to the cell walls, oxidation, and release of intracellular compounds such as proteins, DNA, and lipids.

In this article, we examined the antibacterial effect of two types of discharges: PDD and PDD-driven Ar-PJ. This mode seems promising from the point of view of processing water-containing objects and from the point of view of scaling plasma processing technology. To determine the dominant factor affecting bacteria, we measured the generation of H_2_O_2_, NO_2_−, and NO_3_− in PBS for both operating modes and also monitored the change in surface temperature of the treated samples. 

Plasma produces a variety of active species, including free radicals (e.g., OH•, O•, and N•), ions, and excited molecules. These particles play a significant role in initiating chemical reactions within fluids and biological material [47]. ROS and RNS formed in the plasma–gas phase (1)–(4) can be determined using optical emission spectroscopy.
N_2_ + e^−^ → N• + N• + e^−^, (1)
O_2_ + e^−^ → O• + O• + e^−^, (2)
H_2_O + e^−^ → H• + OH• + e^−^, (3)
Ar + e^−^ → Ar• + e^−^. (4)

The emission spectra for all modes of operation of the “CAPKO” source were previously studied [45]. The spectra consist of systems of N_2_ bands, the first negative system N_2_+ of triplets of atomic oxygen O (777.4 and 844.6 nm), and the weak OH band (306–309 nm). In case of operation with flowing gases, Ar and He lines arise. Based on these measurements, the electron temperatures in the discharge channels, as well as the vibrational and rotational temperatures of N_2_, were determined.

ROS and RNS generated in a multiphase plasma–gas–surface–liquid system, as well as the flow of electrons and UV radiation, trigger a cascade of chemical reactions, as a result of which long-lived compounds are formed in the liquid. Some of the most important from the point of view of biological activity and, at the same time, simple to determine are H_2_O_2_, NO_2_−, and NO_3_−. Therefore, the registration of these compounds often acts as a means of characterizing plasma sources and a measure of their biological activity.

The PDD mode and PDD-driven argon plasma jet used in our study are ineffective from the point of view of ozone generation. Published studies [59] show that ozone generation strongly depends on the power input into the discharge and the gas temperature. In the modes we used in the manuscript, these parameters are relatively high. The “gas” temperature in the discharge channel, measured by spectrometric methods, reached 1500 K [45]. This sharply reduces the concentration of generated ozone. Thus, the main reactive species in the plasma–gas phase are nitrogen oxides and OH radicals.

PDD and PDD-driven Ar-Pj show competitive energy efficiency (energy input per milliliter of liquid, J/mL) compared to other methods of generating low-temperature plasma [15] in relation to H_2_O_2_, NO_2_− and NO_3_− generation. Both modes showed good antibacterial effectiveness. When processed in the PDD mode, local areas of overheating may appear on the surface of the agar, which leads to a change in the color of the agar, and in the case of exposure to biological objects, this can cause their thermal destruction. As the duration of exposure increases, the probability of such areas occurring increases. In the case of Ar-PJ, no overheating of the surface occurs. However, the spark discharge affects only a surface layer of liquid several hundred micrometers thick; that is, a destructive thermal effect on biological objects covered with a sufficiently thick layer of liquid does not occur.

Thus, the PDD mode is not suitable for direct processing of biological objects, and we use it to create plasma-activated liquids or to affect biological objects immersed in liquid. When processing liquids in the PDD mode and when processing any samples in the Ar-PJ mode, the dominant decontamination factor is reactive oxygen species (ROS).

Plasma sources that are widely used in biological applications have their own advantages and disadvantages, which determine the niche they occupy. The advantages of PDD include energy efficiency, compactness, and the ability to create a power supply operating at a relatively low voltage. Also, a piezotransformer, which is an electrode, makes it possible to implement a multifunctional working device that can change the type of generated discharge on the fly. Disadvantages include the sensitivity of the piezotransformer to mechanical influences and the need for cooling to maintain the operating point.

Generally, treatment with CAP is an effective means of influencing gaseous or liquid media and the surfaces of objects of various natures. However, not all types of CAP sources allow processing to be easily scaled. An increase in the processing area of spark discharges, which include PDD, as well as plasma jets, is achieved by multiplying the number of active electrodes [47,60]. This approach is described in the literature both in relation to the preparation of PTS and for direct surface treatment.

The “CAPKO” source also allows us to organize working devices in an array, as well as be placed above the conveyor belt for even greater scaling. The appearance of such a setup is shown in Figure 8.

## 5. Conclusions

Cold atmospheric plasma treatment can be considered an approach to the bacterial decontamination of surfaces of various natures. This is due to its ability to induce oxidative stress, disrupt cellular structures, and modulate cellular processes. In the article, we used the “CAPKO” plasma source to evaluate its ability to deactivate *E. coli* bacteria. Two modes of plasma generation were considered: piezoelectric direct discharge (PDD) and argon plasma jet (Ar-PJ).

We used phosphate-buffered saline and agar as models of contaminated water-containing objects. The first model allows us to estimate the production of reactive oxygen (ROS) and nitrogen species (RNS), which are the main agents of influence on biological objects and can also be used as reference characteristics of the plasma source when reproducing experiments in other laboratory conditions. The second model made it possible to determine the effective impact area and demonstrate the differences in treatment modes in terms of potential thermal damage to the surface of biological objects.

In general, the PDD mode demonstrates better efficiency in the production of ROS and RNS, as well as a larger processing area, which allows it to be used to create plasma-treated solutions, that is, for the indirect effect of CAP on biological objects. This mode has a more random character due to the active movement of the spark channel and therefore the processing spot on the surface. Ar-PJ is characterized by a more even but smaller processing spot. However, it is a safer mode for biological objects from the point of view of thermal damage to the surface.

The scaling of processing in both modes can be carried out by placing a system of several working devices in the form of an array. This approach is a logical and important step for introducing direct and indirect treatment using PDD CAP into widespread practice and is an avenue for future research.

## Figures and Tables

**Figure 1 biomolecules-14-00181-f001:**
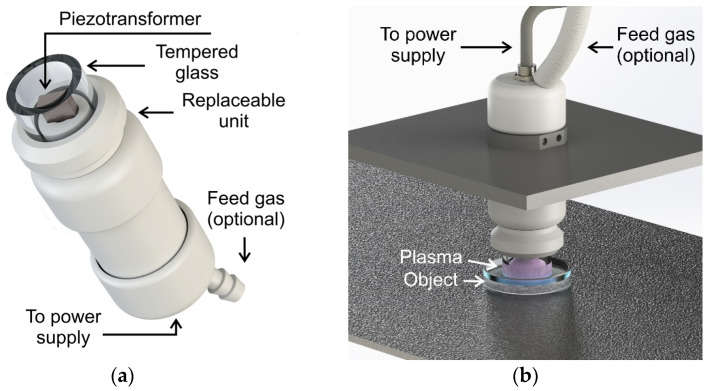
(**a**) Appearance of the working device of the “CAPKO” plasma source and (**b**) treatment setup in the piezoelectric direct discharge (PDD) mode.

**Figure 2 biomolecules-14-00181-f002:**
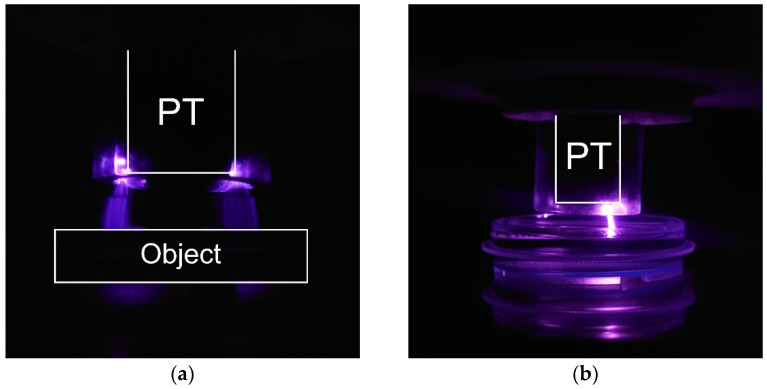
Photo of discharges at the output of a piezotransformer (PT) (**a**) in the corona discharge mode and (**b**) in piezoelectric direct discharge (PDD) mode.

**Figure 3 biomolecules-14-00181-f003:**
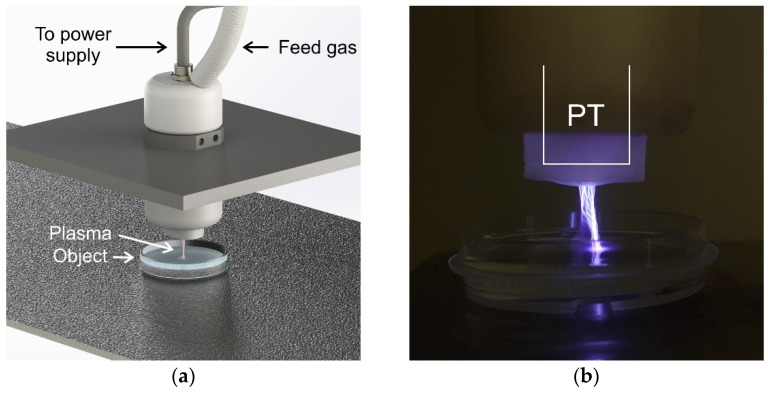
(**a**) Treatment setup in the argon plasma jet mode and (**b**) argon plasma jet photo. PT—piezotransformer.

**Figure 4 biomolecules-14-00181-f004:**
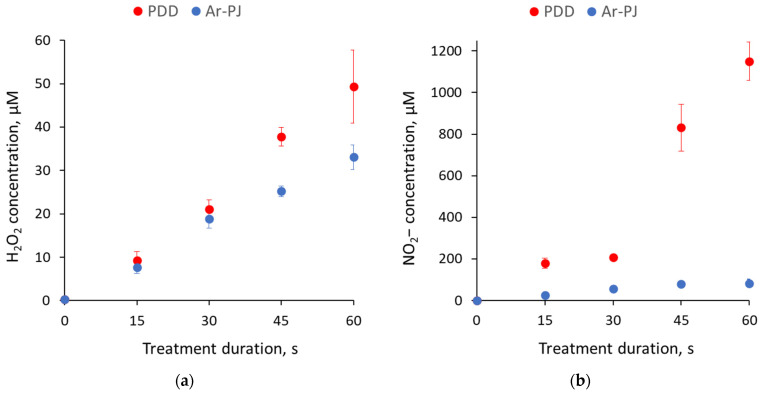
(**a**) H_2_O_2_ and (**b**) NO_2_− concentrations in phosphate-buffered saline treated with piezoelectric direct discharge (PDD) and argon plasma jet (Ar-PJ). Results are presented as mean and standard error.

**Figure 5 biomolecules-14-00181-f005:**
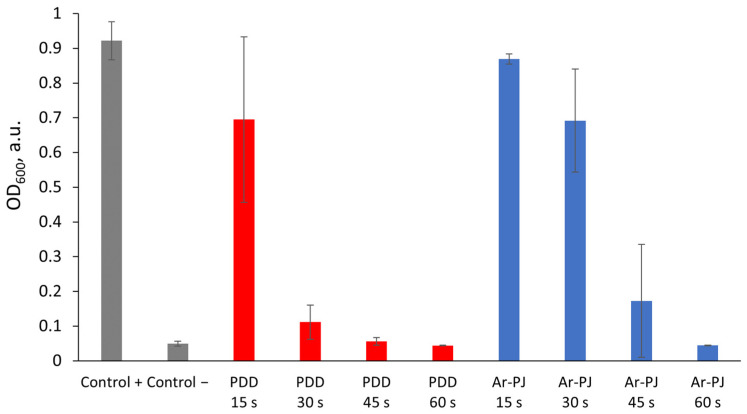
Optical density (OD_600_) of cell suspension in various modes of plasma treatment. PDD—piezoelectric direct discharge; Ar-PJ—argon plasma jet. Results are presented as mean and standard error.

**Figure 6 biomolecules-14-00181-f006:**
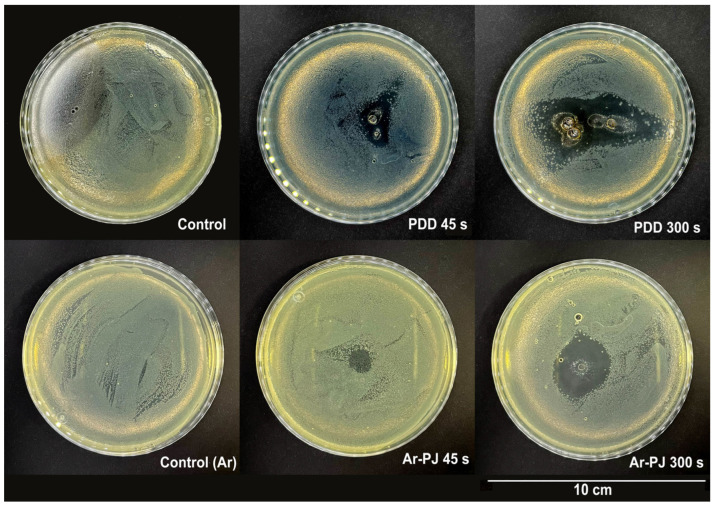
Effective area of bactericidal action of plasma treatment in piezoelectric direct discharge (PDD) mode and argon plasma jet (Ar-PJ) mode for different treatment durations. In the photos, there are 90 mm Petri dishes with agar on the surface, of which a 100 μL of PBS with *E. coli* suspension is applied 24 h of cultivation.

**Figure 7 biomolecules-14-00181-f007:**
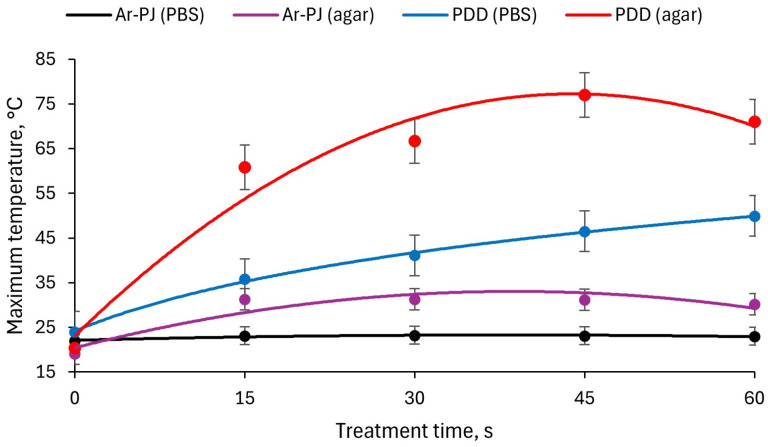
The time dependence of the local maximum temperature (hot spot) on the surface of the sample during treatment. Results are presented as mean and standard error.

**Figure 8 biomolecules-14-00181-f008:**
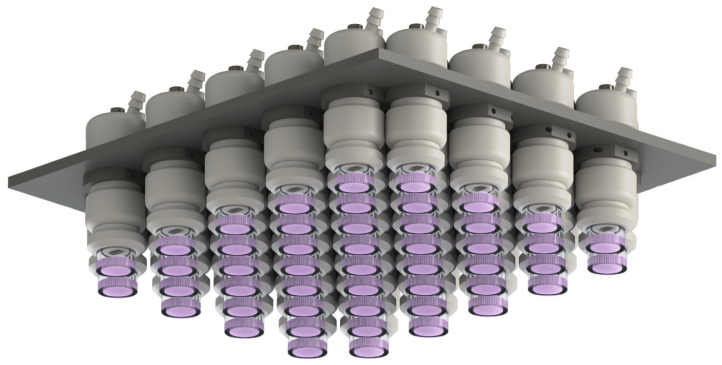
Array system of working devices of “CAPKO” plasma source for treatment scaling.

**Table 1 biomolecules-14-00181-t001:** Experimental setup: plasma treatment modes. CAP—cold atmospheric plasma; PDD—piezoelectric direct discharge; Ar-PJ—argon plasma jet; PBS—phosphate-buffered saline.

CAP Type	Contaminated Object	Treatment Duration, s
PDD	PBS	15
	PBS	30
	PBS	45
	PBS	60
	Agar	45
	Agar	300
Ar-PJ	PBS	15
	PBS	30
	PBS	45
	PBS	60
	Agar	45
	Agar	300

## Data Availability

The data presented in this study are available on request from the corresponding author.

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
