# Peer review of "Bacterial Decontamination of Water-Containing Objects Using Piezoelectric Direct Discharge Plasma and Plasma Jet"

_biomolecules, 2024, doi:10.3390/biom14020181_

Round 1

Reviewer 1 Report

Comments and Suggestions for Authors

Overall, this manuscript is well written and reads smoothly. However, it does not provide more innovative information compared to what is already known in plasma medicine and the authors' previous studies. The ability of cold atmospheric plasmas to produce hydrogen peroxide and nitrite/nitrate in PBS, as well as the ability to kill bacteria, are widely recognized in the plasma medicine community. The results of this paper do not reveal new phenomena or propose new mechanisms, but more of a validation of the effects of an existing device.

 1. In the Introduction section, the authors mention that "However, the mechanisms of action on biological objects remain insufficiently clear due to the complex plasma- chemical and biochemical processes that occur. chemical and biochemical processes that occur in the multi-phase plasma-air-liquid-cell system." However, the work in this paper does not appear to contribute directly to solving this question. The authors may need to consider re-stating what the motivation for this work is.

2. In the Discussion section, a large portion of the section is based on the authors' previous research. There is little discussion of the results from this work and the mechanisms involved. In the revised manuscript, the authors need to adjust and focus the discussion more on the results from this work.

3. It would be more convenient for the readers to give the full name of CAPKO.

4. The phrase "... allows you to ..." is used in several places in the manuscript. It is not very common to use the second person in scientific and technical papers, so the authors are suggested to make revisions.

5. In the experiments carried out in this work, is the power of the PDD mode and the jet mode the same, and what are they? The authors are suggested to provide relevant information, which is helpful to understand the bactericidal efficiency of different modes.

Author Response

Reviewer 1

Comment 1.1

In the Introduction section, the authors mention that "However, the mechanisms of action on biological objects remain insufficiently clear due to the complex plasma- chemical and biochemical processes that occur. chemical and biochemical processes that occur in the multi-phase plasma-air-liquid-cell system." However, the work in this paper does not appear to contribute directly to solving this question. The authors may need to consider re-stating what the motivation for this work is.

Reply 1.1

The Introduction section has been revised to more clearly describe the purpose of the research conducted.

Comment 1.2

In the Discussion section, a large portion of the section is based on the authors' previous research. There is little discussion of the results from this work and the mechanisms involved. In the revised manuscript, the authors need to adjust and focus the discussion more on the results from this work.

Reply 1.2

We have moved the description of previous studies to the Introduction section. In the Discussion section we focused on discussing the results from this work.

Comment 1.3

It would be more convenient for the readers to give the full name of CAPKO.

Reply 1.3

 “CAPKO” is the name of the device model and it does not have a public expanded version. When creating the name, we used the abbreviation CAP (cold atmospheric plasma) and the first two letters of the name of the author of the concept.

Comment 1.4

The phrase "... allows you to ..." is used in several places in the manuscript. It is not very common to use the second person in scientific and technical papers, so the authors are suggested to make revisions.

Reply 1.4

Corresponding changes have been made to the text.

Comment 1.5

In the experiments carried out in this work, is the power of the PDD mode and the jet mode the same, and what are they? The authors are suggested to provide relevant information, which is helpful to understand the bactericidal efficiency of different modes.

Reply 1.5

The power consumption of the piezotransformer did not change in the experiments in all modes and amounted to 3 W. The average power supplied to the discharge was approximately 1.5 W. This power is distributed during the processing between all spark discharges with a duration of ~10 ns, following with a frequency of ~42 kHz (approximately one spark per half-cycle of the supply voltage). In the Ar-PJ mode due to the use of an argon flow, processing is possible with a larger distance between the electrode and the object. In addition, the gas flow cools the surface. This significantly affects the chemical reactions occurring in the system.

We have included this explanation in the Methods section.

Reviewer 2 Report

Comments and Suggestions for Authors

The paper ‘Bacterial decontamination of water-containing objects using piezoelectric direct discharge plasma and plasma jet’ deals with the use of a piezoelectric plasma device used for decontamination purposes. Different device configuration and treated substrate have been investigated. Results show the decontamination effects of this power supply used with air and as plasma jet utilizing argon as carrier gas. It is my opinion that the paper is clear and well structured. Despite this, some data are missing (especially temperature increment in the treated object) and a clear improvement in the decontamination field, operated by this kind of device, it is not showed. For this reason, I recommend a major revision.

Main concerns are listed below.

·         English should be checked. As an example:

o   Line 61 ‘All methods of objects treatment using CAP can be divided…’. This sentence is not that clear.

o   Line 86 ‘Thus, a comparison was made of the results for PDD plasma and plasma jet with an argon flow, which is important…’. This sentence is not that clear.

·         Author claim that ‘Plasma sources using PDD are primarily characterized by energy efficiency, increased electrical safety during operation, and flexibility in the design of the working (output) device, which directly acts as a high-voltage electrode that creates plasma.’ In reference 43 an 45 it is mentioned that the ozone production rate is of 73 mg/h for and average power of 8 W, this means about 9 mg/Wh. Commercial ozone generator usually own rates in the order of 50-60 mg/Wh. How can authors claim that their source is efficient in ozone production? Why is their source electrical safety? As a matter of fact their source presents an high voltage electrode that is exposed. In this way an harmful direct contact with a high voltage terminal could occur.

·         In the two device configurations (PDD and Ar-PJ), which are average power delivered to the discharge? At line 134 it is reported a value of 3 W. It is my opinion that this value I really high if compared with values usually found for plasma jets (below 1 W). This means that the discharge should be quite hot. Authors mention the possibility to induce thermal damage with long treatment time. They should report the temperature of treated sample as a function of time. In Fig. 6, as an example, it seems that there is a thermal damage. This is a crucial point in my opinion, because the treatment becomes thermal.

·         Biological results should be compared with data available in the literature to make a comparison. Why this source is better that other? The paper do not clarify this point.

Comments on the Quality of English Language

English must be slightly improved

Author Response

Reviewer 2

Comment 2.1

English should be checked. As an example:

  • Line 61 ‘All methods of objects treatment using CAP can be divided…’. This sentence is not that clear.
  • Line 86 ‘Thus, a comparison was made of the results for PDD plasma and plasma jet with an argon flow, which is important…’. This sentence is not that clear.

Reply 2.1

The sentences have been revised.

Comment 2.2

Author claim that ‘Plasma sources using PDD are primarily characterized by energy efficiency, increased electrical safety during operation, and flexibility in the design of the working (output) device, which directly acts as a high-voltage electrode that creates plasma.’ In reference 43 an 45 it is mentioned that the ozone production rate is of 73 mg/h for and average power of 8 W, this means about 9 mg/Wh. Commercial ozone generator usually own rates in the order of 50-60 mg/Wh. How can authors claim that their source is efficient in ozone production?

Reply 2.2

PDD mode and PDD-driven argon plasma jet used in our study are ineffective from the point of view of ozone generation. Published studies [10.1016/j.vacuum.2021.110647] show that ozone generation strongly depends on the power input into the discharge and the gas temperature. In the modes we used in the manuscript, these parameters are relatively high. The “gas” temperature in the discharge channel, measured by spectrometric methods, reached 1500 K [10.1007/s11182-020-01948-1]. This sharply reduces the concentration of generated ozone. Thus, the main reactive species in the plasma-gas phase are nitrogen oxides and OH radicals.

We have included this explanation in the Discussion section.

Comment 2.3

Why is their source electrical safety? As a matter of fact their source presents an high voltage electrode that is exposed. In this way an harmful direct contact with a high voltage terminal could occur.

Reply 2.3

By declaring that the source is electrically safe, we meant the fact that high voltage is created at the very end of the electrode (piezotransformer) on the working device. Thus, a voltage of 30-100 V is supplied to the element, which the operator (user) must hold in his hand (and for some piezotransformers, about 10 V [10.1109/TPS.2018.2870345]). This simplifies the use of the device due to the reduction in the number of potentially dangerous elements for contact. This also allows to create a more compact source. However, we agree that for this class of devices in which direct contact with high voltage is possible, the term “electrically safe” was used incorrectly by us, so we excluded it from the text.

Comment 2.4

In the two device configurations (PDD and Ar-PJ), which are average power delivered to the discharge? At line 134 it is reported a value of 3 W. It is my opinion that this value I really high if compared with values usually found for plasma jets (below 1 W). This means that the discharge should be quite hot. Authors mention the possibility to induce thermal damage with long treatment time. They should report the temperature of treated sample as a function of time. In Fig. 6, as an example, it seems that there is a thermal damage. This is a crucial point in my opinion, because the treatment becomes thermal.

Reply 2.4

The power consumption of the piezotransformer did not change in the experiments in all modes and amounted to 3 W. The average power supplied to the discharge was approximately 1.5 W. This power is distributed during the processing between all spark discharges with a duration of ~10 ns, following with a frequency of ~42 kHz (approximately one spark per half-cycle of the supply voltage). The “gas” temperature in the discharge channels in both processing modes is relatively high compared, for example, with a dielectric barrier discharge. We measured this temperature in detail, and we are preparing a comprehensive article for a technical journal. In the previous article [doi:10.1007/s11182-020-01948-1] we present the results of preliminary measurements of electron and ion temperatures in various modes for the “CAPKO” plasma source.

For this manuscript, we measured the surface temperature of processed samples. The results are shown in detail in the attached file. The change in object temperature during processing was determined using an Optris PI 640 infrared camera (Optris, GmbH, Berlin, Germany) and Optris PIX Connect software (Optris, GmbH, Berlin, Germany).

In the case of processing contaminated PBS, the maximum local surface temperature (hot spot) practically does not increase when processing in the Ar-PJ mode. In PDD mode it can reach 50 degrees at the point of contact of the discharge with the surface, but the temperature near this point is much lower.

When treating the surface of contaminated agar, the temperature of the hot spot in the Ar-PJ mode is around 30 degrees throughout the entire duration of treatment. In PDD mode it can reach 80 degrees, due to which the agar melts and takes on a different color (this can be seen in Figure 6 from the manuscript). However, this temperature is achieved in a region of small area; the temperature a few millimeters from this point is significantly lower.

In addition, the spark discharge affects only a surface layer of liquid several hundred micrometers thick, that is, a destructive thermal effect on biological objects covered with a sufficiently thick layer of liquid does not occur.

Thus, PDD mode is not suitable for direct processing of biological objects, and we use it to create plasma-activated liquids, or to affect biological objects immersed in liquid. When processing liquids in the PDD mode and when processing any samples in the Ar-PJ mode, the dominant decontamination factor is reactive oxygen species (ROS).

We have added a corresponding description with figure to the manuscript in the Materials, Results and Discussion sections.

Comment 2.5

Biological results should be compared with data available in the literature to make a comparison. Why this source is better that other? The paper do not clarify this point.

Reply 2.5

Plasma sources that are widely used in biological applications have their own advantages and disadvantages, which determines the niche they occupy. The advantages of PDD include energy efficiency, compactness, and the ability to create a power supply operating at a relatively low voltage. Also, a piezotransformer, which is an electrode, makes it possible to implement a multifunctional working device that can change the type of generated discharge on the fly. Disadvantages include the sensitivity of the piezotransformer to mechanical influences and the need for cooling to maintain the operating point.

We have included this explanation in the Discussion section.

Round 2

Reviewer 1 Report

Comments and Suggestions for Authors

The authors have addressed most of my questions. 

Reviewer 2 Report

Comments and Suggestions for Authors

Authors fulfilled all reviewers' requests. The paper can now be published.